# First Application of the Integrated Karst Aquifer Vulnerability (IKAV) method. Potential and Actual vulnerability in Yucatan, Mexico

Miguel Moreno-Gómez[1], Carolina Martínez-Salvador[2], Rudolf Liedl[3], Catalin Stefan[1] and Julia Pacheco[4]

[1]Research Group INOWAS, Department of Hydrosciences, Technische Universität Dresden, 01069 Dresden, Germany
[2]Engineering Institute, National Autonomous University of Mexico (UNAM), 04510, Mexico City, Mexico
[3]Institute of Groundwater Management, Department of Hydrosciences, Technische Universität Dresden, 01069 Dresden, Germany
[4]Department of Environmental Engineering, Yucatan Autonomous University (Retired), 97203 Merida, Mexico

*Correspondence to*: Catalin Stefan (Catalin.Stefan@tu-dresden.de)

**Abstract.** Groundwater vulnerability maps are important decision support tools for water resources protection against pollution and helpful to minimize environmental damage. However, these tools carry a high subjectivity along the multiple steps taken for the development of such maps. Additionally, the theoretical model on which they are based does not consider other important parameters, such as pollutant concentration or pollutant residence time in a given section of the aquifer, solely focusing on the theoretical travel time of a pollutant particle from a release point towards a target. In this work, an integrated methodology for the evaluation of potential (intrinsic) and actual vulnerability is presented. This integrated method, named IKAV, was developed after the analysis of several study cases around the world and the application of multiple intrinsic groundwater vulnerability methods in a selected study area. Also, a solute transport model served as the basis to define additional parameters for vulnerability analysis for areas severely affected by anthropogenic practices. However, the focus of the model must not be mistaken with hazards and risk mapping. A defined workflow and several criteria for parameters and attributes selection, rating and weighting, and vulnerability classification are here presented. The first application of the IKAV method was carried out in the Yucatan karst, demonstrating to be a reliable method for vulnerability estimation. Results demonstrated the scope of the IKAV method in highlighting important regional conditions, minimizing the subjectivity, and expanding the analysis of vulnerability.

## 1 Introduction

Since the introduction of the groundwater vulnerability concept by Albinet (1970), several redefinitions and subclassifications for this concept have been continuously proposed. The current groundwater vulnerability conceptual model evaluates the geological, hydrological, and hydrogeological characteristics of a given area by their ability to transport a pollutant particle from a release point towards a specific location in the aquifer (Zwahlen, 2003). The general process to estimate this sectional aquifer vulnerability then follows a release-pathway-target model (Goldscheider, 2005). With focus on the natural

characteristics along the pathway, such as soils, vegetation, lithology, and slope, among others, the travel time of a theoretical and immutable pollutant particle is estimated. This "intrinsic" (natural) vulnerability does not consider the possible changes that a pollutant particle can experience along the pathway; when such changes are also evaluated, the vulnerability analysis is then referred to as "specific" (Vrba and Zaporožec, 1994). However, the extensive cost to obtain data and carry out field work

to evaluate the possible changes of a given pollutant along the pathway, turn specific vulnerability analysis less applicable in comparison with intrinsic vulnerability maps.

Given the high heterogeneity and anisotropy of karst, groundwater vulnerability methods for porous aquifers are not efficient to estimate karst aquifer vulnerability. According to literature, EPIK is the first proposed method to evaluate vulnerability of karst aquifers with the aim to define protection areas and to fulfil Swiss water regulations (Dörfliger et al., 1999). The EPIK,

the DRASTIC (Aller et al., 1987), and the European framework (Zwahlen, 2003) are the basis for multiple vulnerability methods for karst that have arisen during the last two decades (Iván and Mádl-Szőnyi, 2017; Parise et al., 2018). The recurrent appearance of new methods and/or adaptations to estimate groundwater vulnerability in karst exhibits the complexity of such systems. The existence of multiple methodologies following the same purpose complicates the selection of an appropriate method to be applied for a given karst area. Application of several methods over the same region usually displays a considerable

mismatch among final vulnerability classifications (Gogu et al., 2003; Ravbar and Goldscheider, 2009; Marín et al., 2012; Stevanović, 2015; Kavousi et al., 2018). However, recent studies have demonstrated that, under some regional conditions, outcomes from multiple vulnerability methods can display a remarkable agreement in vulnerability classification; nevertheless, a high correlation among methods must never be taken as a confirmation for reliability (Moreno-Gómez, 2021).

In general, two main problems are directly related to current methodologies to estimate intrinsic vulnerability: the high

subjectivity and the uncertainty of the current conceptual model. For the former, several factors influence the high subjectivity along the steps to estimate vulnerability such as personal interpretations, the use of dissimilar standards, inconsistent parameters and attributes, and discretional rates and weights (Figure 1). Regarding the latter, the sole evaluation of the theoretical travel time of an immutable pollutant particle can severely mislead the vulnerability analysis and decisions based on it; this problematic has been pointed out by the COST Action 620 final report (Zwahlen, 2003). Therefore, the reliability

of the intrinsic vulnerability analysis is arguable for karst aquifers that are already affected by anthropogenic activities.

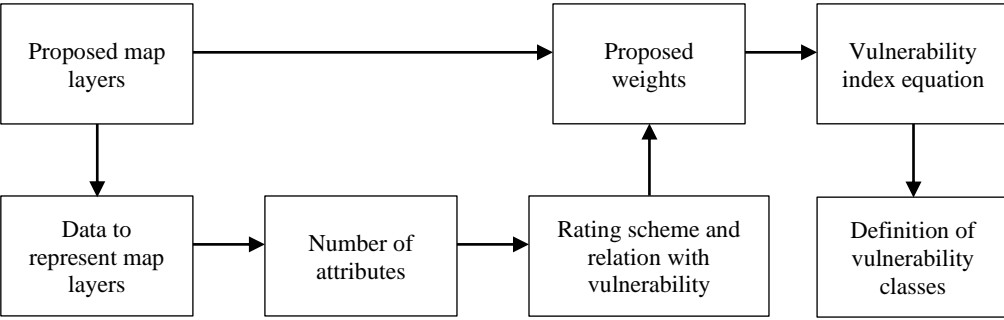

**Figure 1: Steps subject to discrepancy from Point Count System Methods (PCSM) evaluating vulnerability.**

Given the high heterogeneity of karst, either locally, regionally, or continentally, standardization of parameters (including attributes, rates, and weights) utilized to estimate intrinsic groundwater vulnerability is a very complex task (Ford and Williams, 2007; Parise et al., 2015). However, a standardized process can help to minimize the subjectivity of the current conceptual model. From previous studies with the aim to develop an integrated karst aquifer vulnerability approach, important considerations have been highlighted for maps selection, parameters filtering, attributes discretization, and rating schemes (Moreno-Gómez et al., 2018, 2019; Martínez-Salvador et al., 2019; Moreno-Gómez, 2021).

Nowadays, the anthropogenic stress that karst aquifers around the world are experiencing is undeniable. Pollution generated by agricultural practices, cattle raising, wastewater disposal, and dumping sites, among others, have already affected karst groundwater quality in some regions (Parise et al., 2004; Pronk et al., 2007; Ravbar and Kovacic, 2015). On the other hand, large wellfields, with considerable water extraction volumes, can diverge groundwater flow from its natural course. This increases the uncertainty of current intrinsic vulnerability methods (either for source or resource estimations) with regard to their application on already anthropogenically-affected karst areas. In order to evaluate current vulnerability scenarios, solute transport models can be beneficial additional criteria to enhance the role of vulnerability maps as decision support tools (Martínez-Salvador, 2018; Martínez-Salvador et al., 2019).

With the aim of solving the aforementioned problems and enhancing the scope of groundwater vulnerability as a decision support tool, the Integrated Karst Aquifer Vulnerability (IKAV) method is here presented as the groundwork for further vulnerability studies. This integrated strategy was developed based on three research steps: 1) the detailed and critical analysis of numerous applications of current intrinsic vulnerability methods around the world; 2) the application and comparison of eight selected methods on a karst study area; and 3) the application and evaluation of a transport model in a severely polluted karst region to determine additional criteria for vulnerability estimation. The state of Yucatan, an important karst region located in Mexico, was selected to carry out steps 2 and 3. The IKAV method demonstrates the necessity to expand the vulnerability analysis beyond the 2-dimentional "intrinsic" scheme and the importance of considering the current vulnerability scenario of the evaluated aquifer. This work presents the first application of IKAV; the study area is the Mexican state of Yucatan.

**2 Study area**

The state of Yucatan (39,500 km$^2$) is located in the northern part of the Yucatan Peninsula, a transboundary limestone platform (approximately 160,000 km$^2$) covering parts of Mexico, Guatemala, and Belize (Figure 2a). According to Weidie (1985), the Peninsula is formed by limestone, dolomites, and evaporites with thicknesses reaching up to 1,500 m. Rocks at the surface correspond to northward sequences from the Upper Cretaceous (Paleogene period) to Holocene (Quaternary period) epochs (Butterlin, 1958; Bonet and Butterlin, 1962; López-Ramos, 1975). The Yucatan Peninsula is classified as a well-developed karst given the existence of systems of solutional conduits of considerable diameter in the sub-surface, extending in the range of kilometres (Bauer-Gottwein et al., 2011). The Yucatan state (hereinafter Yucatan) has four hydrogeological regions: Coastal Area, Inner Cenote Ring, Central Plain, and Valleys and Hills. Yucatan presents interesting characteristics such as high doline

(cenote) density areas, regional faults, and a nearly flat topography for most of the state (Figure 2b); other important characteristics of Yucatan, such as the shallow water table, soil distribution, the spatial precipitation pattern, and lithology are visually displayed in Appendix A.

The nearly flat topography and the considerable secondary porosity do not allow surface streams to generate, making diffuse infiltration dominant in Yucatan (Figure 2c). For a more detailed information regarding the characteristics of Yucatan, the

works of Bonet and Butterlin (1962), Doehring and Butler (1974), Lesser (1976), Pope et al. (1991), (1993), Hildebrand et al. (1995), and Lugo-Hubp and García (1999) are highly recommended.

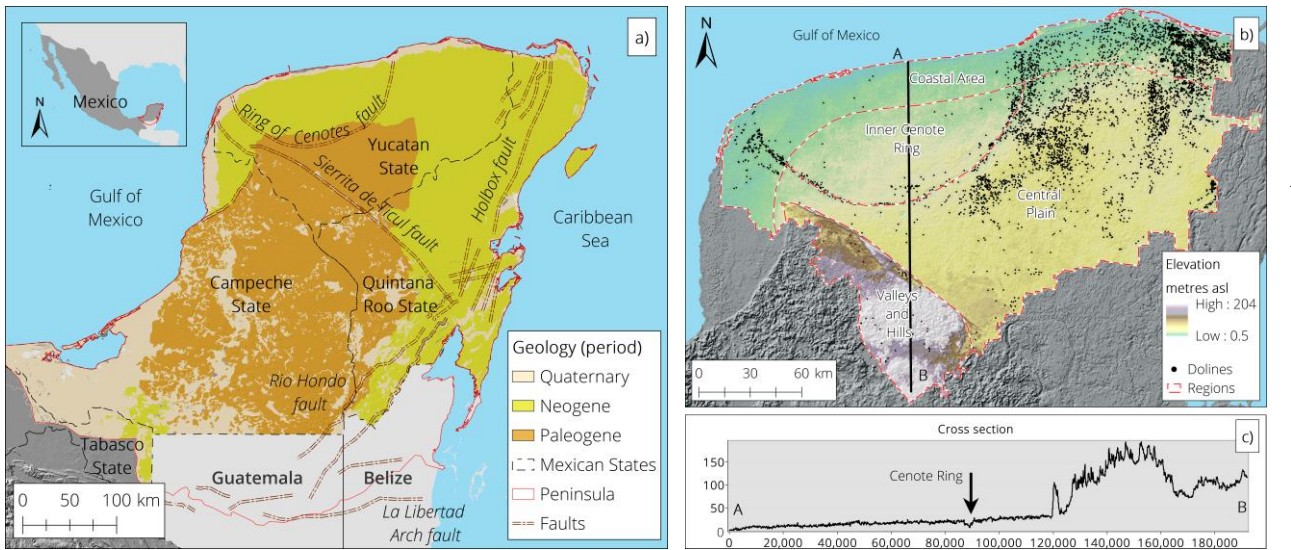

**Figure 2: The Yucatan Peninsula's characteristics. In a) the Peninsula limits, geology, and regional faults; in b) Yucatan's elevation and dolines location; in c) a cross section of Yucatan displaying the nearly flat topography.**

Yucatan is divided into 106 municipalities of variable extension and population density (Figure 3a). An important area in Yucatan is the Merida Metropolitan area (MMA), located in the Inner Cenote Ring region and composed by six municipalities (Figure 3b). This highly urbanized area with around 1.1 million inhabitants (approximately 52% of Yucatan's population) presents several environmental problems related to fast urbanization and the practices derived from it (Pacheco and Cabrera, 1997; Marín et al., 2000; Pacheco et al., 2002; González-Herrera et al., 2014; Rojas-Fabro et al., 2015). It is estimated that

more than 55% of the water extraction for human consumption takes place in the MMA (Figure 3c). Wastewater volumes are estimated as 75 to 80% of the extracted water for human consumption. With approximately 80% of the population of the MMA utilizing artisanal septic tanks, which are permeable, a pollution plume has been generated from the continuous leaking of waste water towards the aquifer(Graniel et al., 1999; Marín et al., 2000). The pollution plume has turned the upper 20 metres of the aquifer below the city of Merida unsuitable for human consumption (L. Marín, personal communication, July 2017).

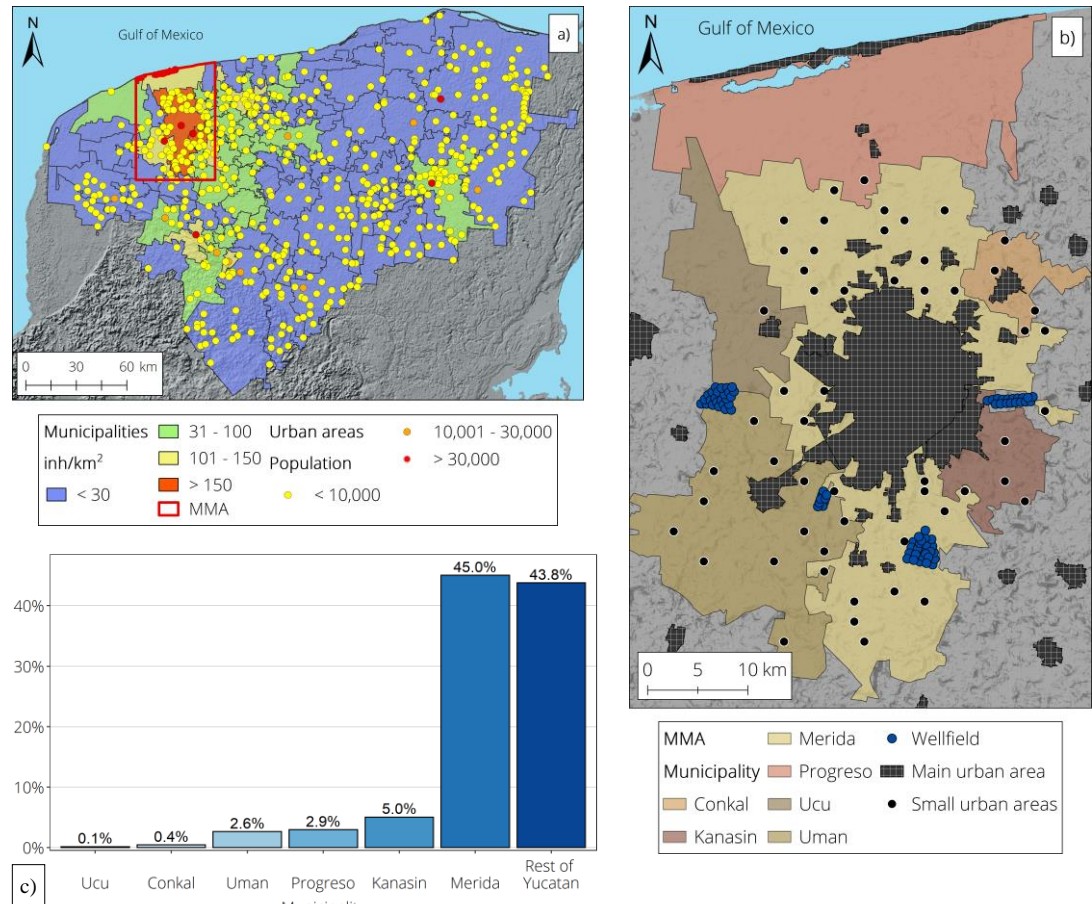

**Figure 3: Administrative and anthropogenic characteristics in Yucatan. In a), the hydrogeological division and population density by municipality; in b), the MMA and the main urban areas; in c), comparative of water consumption for the MMA and the rest of Yucatan.**

The continuous disposal of untreated waste water into the aquifer, the use of nitrogen-rich fertilizers used in agriculture, and large-scale pig farming has put Yucatan under a severe nitrate ($NO_3^-$) pollution scenario (Pacheco and Cabrera, 1997; Pacheco et al., 2001, 2002; Drucker et al., 2003; Pacheco-Ávila et al., 2004b; Delgado et al., 2010). Given that intrinsic vulnerability solely evaluates the probable advection of a theoretical pollutant, no other characteristics reflecting the current vulnerable state of the aquifer (or water extraction wells) are analysed. In the Yucatan karst, as well as many other already affected karst areas, the inclusion of additional parameters for vulnerability estimation is critical.

## 3 The IKAV method

The proposed IKAV method aims to minimize the subjectivity of the current process to estimate intrinsic resource vulnerability and to provide additional parameters, such as pollutant concentration from solute transport, to enhance the vulnerability analysis. Given that intrinsic vulnerability evaluates an "IF" condition (vulnerability under an incidental pollution scenario)

while solute transport aims to simulate a current vulnerable scenario, IKAV is presented as a complementary analysis of these two conditions; potential vulnerability (IKAV-P) and actual vulnerability (IKAV-A) maps are proposed. IKAV is based on three interconnected principles to be fulfilled: infiltration distinctive, regionalization, and representative activity/pollutant (Figure 4).

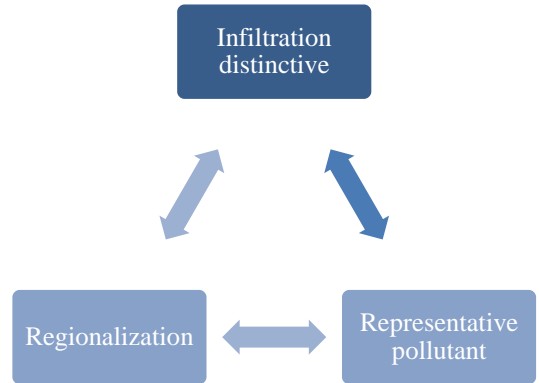

Figure 4: The proposed principles for an integrated evaluation of vulnerability.

In the infiltration distinctive principle, the objective is to define the goal of the analysis, settling either a point infiltration or diffuse infiltration condition; this selection dictates how some parameters will be evaluated. For example, if a point infiltration scenario is going to be investigated, high slopes and fine-textured soils will represent a more vulnerable condition due to their runoff generation capacity; however, these intrinsic characteristics will have an opposite role in vulnerability for diffuse infiltration conditions.

Regionalization settles the individual ranges for the selected parameters; here it is critical how the rates will be assigned to attributes in congruence with the criteria from the infiltration distinctive procedure. This step aims to evaluate vulnerability according to existing conditions in a given study area, avoiding the indirect evaluation of external characteristics proposed by current methodologies but not present in the area of interest. The link of this principle with the infiltration distinctive principle relies on the rating scheme of some parameters and their further influence to be represented by weights.

The representative pollution principle defines the contaminant to be evaluated in correspondence with anthropogenic activities carried out in a given study area. The representative pollutant, or the activity from which it is derived, is directly linked to the infiltration condition. For example, the use of fertilizer in hillside farming could represent a more vulnerable condition for a swallow hole catchment than sewage leakage occurring in the same area.

In addition of these principles, a group of multiple criteria is proposed in order to minimize the high subjectivity and the discretional definition of vulnerability classes for the IKAV-P. This additional criterion is also beneficial for the establishment of vulnerability classes for the IKAV-A, according to the permissible maximum of the studied pollutant or the purpose of the extracted water at a given point of the aquifer (e.g., human consumption or irrigation).

IKAV-P is proposed as a multi-criteria decision analysis (MCDA) following a rating-weighting system. As proposed by several authors, this type of analysis must evaluate attributes following certain characteristics for their classification (Carver, 1991; Heywood et al., 1995; Malczewski, 1999, 2006). Therefore, with focus on karst vulnerability, the attributes must be:

160

• Measurable – Given the fact that it is challenging to measure some attributes in karst areas, such as the degree of karstification, statistical tools can help to better define notable differences in the study area.

• Non redundant – According to current intrinsic vulnerability methods, some parameters can be defined utilizing the same base map (e.g., karstification and epikarst development from karst surface features according to EPIK). Inclusion of multiple parameters derived from the same evaluation process must be avoided to prevent over/under vulnerability estimations.

• Minimal – Considering that some degree of subjectivity is inherently attached to each evaluated characteristic, the use of an extensive number of parameters can complicate the process. Therefore, the number of evaluated parameters must be kept as minimal as possible, only considering the relevant ones to characterize and evaluate specific karst features.

• Availability – the objective must be achieved with the available data from the region. Under conditions of low-resolution data, such data must be excluded, even if it has a significant influence on the objective.

From the analysis of multiple study cases around the world and the application of eight intrinsic groundwater vulnerability methodologies in the Yucatan karst (see Moreno-Gómez et al., 2018, 2019), it was possible to analyse and redefine additional criteria to be added to the previous aspects. In order to improve the intrinsic vulnerability analysis with the IKAV-P, the following attributes' criteria must also be considered:

• Variable - Given that the goal of potential intrinsic vulnerability (or any other MCDA) is to provide a range of conditions to evaluate their relevance for a given objective, a map layer displaying homogeneity will have no purpose for the research. Although a homogeneous map layer could be useful when comparing vulnerability maps of different areas, homogeneous layers must be excluded from the analysis to achieve a regionalization principle.

• Unambiguous – The objective must be clear and well defined. Either for point or diffuse infiltration, the characteristics influencing such processes must be considered separately; this means the assessing performance of an individual analysis for each scenario, avoiding the use of "rest of the area" conditions.

• Distinctive - The influence of some parameters is different for point and diffuse infiltration, therefore, rates and weights

assigned to individual parameters must be evaluated according to the objective. This is clear for slope and soil texture, the influence of which must be evaluated as oppositional for each scenario.

• Territorial – The number of attributes, their evaluation, and rating must be solely dependent upon the local or regional characteristics but following a rating pattern linked to the objective. Attributes proposed by other methods but not present in

the study area must be avoided as vulnerability indicators.

IKAV-A is proposed based on solute transport modelling to represent an approximation of a current pollution scenario. By taking the most influential anthropogenic activities, a representative pollutant can be utilized for a better interpretation of resource (or source) vulnerability by pollution levels (permissible maximum). Activities affecting groundwater quality in karst

areas are variable, as well as the pollutants derived from such activities. Therefore, several IKAV-A maps can be produced according to the regional anthropogenic activities. The definition of actual vulnerability rates will depend on the type of pollutant, regional regulations, and the purpose of a given extraction well. Having defined the three principles of IKAV and a general criterion for map layer selection (including rating and discretization), a workflow is presented as the groundwork to evaluate an integrated groundwater vulnerability (Figure 5).


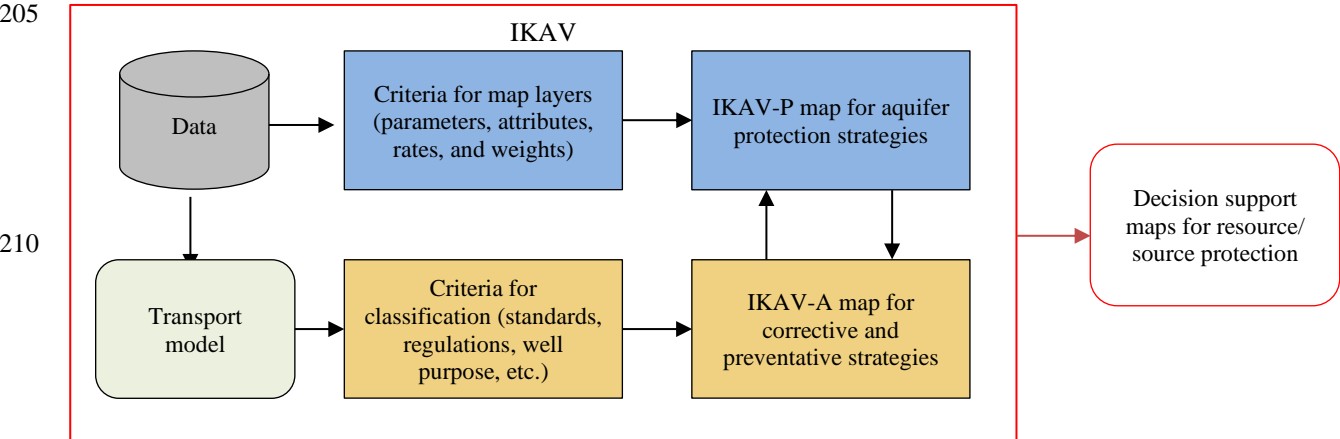

**Figure 5: Summarized workflow for the estimation of an integrated vulnerability (potential and actual vulnerability). Modified after Moreno-Gómez (2021).**

The process is considered as an improved guideline to obtain a potential (intrinsic) groundwater vulnerability map with the addition of an actual vulnerability map from solute transport based on the regional anthropogenic activities. For IKAV-P, the data utilized to generate the necessary map layers was obtained from public sources (see Appendix B). For IKAV-A, a solute

transport model of the MMA, presented by Martínez-Salvador et al. (2019), was utilized with slight modifications.

## 4 Application of the IKAV

### 4.1 IKAV-P

Following the previously presented criteria in order to fulfil the IKAV principles, IKAV-P was tested in Yucatan. Given the Yucatan hydrogeological characteristics, the infiltration scenario was regionally defined as diffuse due to the low relief of the
area and the high fissuring conditions not allowing surface streams to generate. Therefore, the vulnerability analysis and the rating of attributes strictly followed a vertical advection point of view. A filtering process for the available data, according to the attribute's selection criteria, helped to exclude parameters not contributing to the analysis. Yucatan's lithology is considered to be homogeneous (limestone) at a regional scale; therefore, no significant differences can be evaluated for our purposes, with the lithology map excluded from the process. Having defined a regionally diffuse infiltration condition, high slopes represent
the most protective attribute from the topographic parameter. Nevertheless, given the low variability displayed by the slope map (see appendix A) the slope parameter was also excluded. Vegetation is commonly associated with infiltration conditions and its influence on runoff generation (see EPIK, COP, PI, and the Slovene approach). Having defined a diffuse infiltration condition for our study area, vegetation maps were excluded from the analysis. Similarly, due to the lack of a data regarding the spatial distribution of recharge, this parameter was also excluded from the IKAV-P but included as a boundary condition
in the IKAV-A.

Given the lack of indirect data in Yucatan, such as spring flow, doline density maps can be used as representative of karstification, epikarst, or aquifer development. However, given the fact that the use of three map layers derived from doline density could lead to over-/under-estimations, doline density was selected to generally represent a karstification map (Figure 6a); fissuring density maps were excluded from the analysis due to their similar discretization with those from doline density
(see Appendix A). Given the contrasting depths at which groundwater can be found in Yucatan, the thickness of the unsaturated zone was selected as a vulnerability parameter. The attributes discretization of this parameter was carried out to highlight the shallowness of the water table in the flat plain and its deeper location in the Valleys and Hills region (Figure 6b). Soils were evaluated based on clay content percentage for the relationship between hydraulic conductivity (K) and fine particles content in soils (Figure 6c). This evaluation of soils as vulnerability parameter is more sensible and serves to avoid the subjectivity
derived from the multiple standards commonly applied to classify vulnerability from soil texture. For this work, it was decided not to include soil thickness as part of the analysis, given the extension of the study area and the low spatial resolution of data from boreholes. Arbitrarily, an inverse relationship was defined for rating purposes: the lower the rating, the higher the vulnerability.

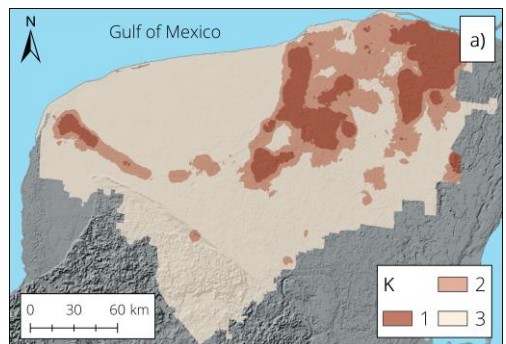
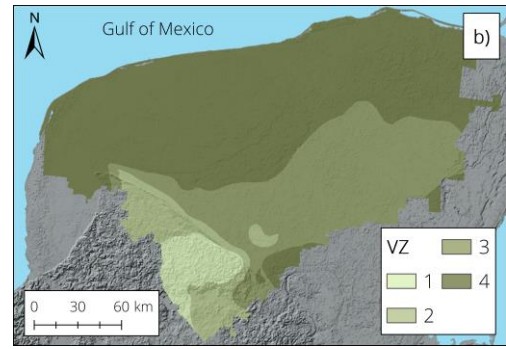
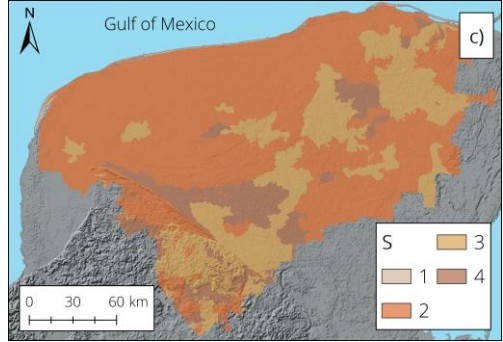

**Figure 6: Selected map layers and attributes' rates according to the proposed criteria for IKAV-P. In a), doline density as representative of karstification (K); in b), depth to groundwater as representative of the vadose zone (VZ); in c), soil clay content as representative of the overlying protection (S).**

In total, three parameters, representing a notable variability, were selected from the data filtering step to represent karstification, vadose zone, and soils. In order to avoid misrepresentations from the usually applied discretional approach and to fulfil the regionalization rule, the attributes classification was performed statistically. The Jenks classification method (natural breaks) was utilized for this purpose; this method minimizes the variance within a given attribute and maximize the variance between them (Jenks, 1967). Parameters' attributes were rated solely in reference to vertical advection (infiltration distinctive rule) allowing pollutants to migrate from the surface to the water table. The ranges assigned by the Jenks classification intent to fulfil the regionalization rule; hence, these ranges are only applicable for our study area (Table 1).

**Table 1: Selected map layers after the criteria filtering process proposed by the IKAV.**

| Map layers | Parameters | | Attributes | | | |
|---|---|---|---|---|---|---|
| **Karstification** | Doline density | Range | > 4 per km² | 2 - 4 per km² | < 2 per km² | - |
| **(K)** | | Rating | 1 | 2 | 3 | - |
| **Vadose zone** | Depth to | Range | < 20 m | 21 - 30 m | 31 – 100 m | > 100 m |
| **(VZ)** | groundwater | Rating | 1 | 2 | 3 | 4 |
| **Soils (S)** | Clay content | Range | <15% | 16 - 30% | 31 - 40% | > 40% |
| | | Rating | 1 | 2 | 3 | 4 |




Regionally, the most vulnerable attributes are: high doline density areas, a shallow water table, and soils with low clay content. Previous tests of IKAV-P indicate that a correlative rating system is beneficial for a statistical classification of the vulnerability index and determine vulnerability classes. After the classification of attributes and the assignment of rates, a procedure to define the importance of each map layer was carried out. The analytical hierarchy process (Saaty, 2008), one of the several methods utilized for MCDA, was selected to determine the ranking or importance of each parameter for IKAV-P. After the standardization of the pair-wise comparison matrix, the eigen values (priority) were obtained (Table 2).

**Table 2: Analytical hierarchy process (AHP) and eigenvalues for the selected map layers.**

| **IKAV-P AHP** | | | |
|---|---|---|---|
| | **K** | **VZ** | **S** |
| **K** | 1 | 1 | 5 |
| **VZ** | 1 | 1 | 3 |
| **S** | 0.2 | 0.33 | 1 |
| **Sum** | 2.2 | 2.33 | 9 |

| **IKAV-P AHP normalized** | | | |
|---|---|---|---|
| | **K** | **VZ** | **S** | **Priority (weight)** |
| **K** | 0.455 | 0.429 | 0.556 | 0.480 |
| **VZ** | 0.455 | 0.429 | 0.333 | 0.405 |
| **S** | 0.091 | 0.142 | 0.111 | 0.115 |
| **Sum** | 1.00 | 1.00 | 1.00 | 1.00 |

With the consistency index (CI) being lower than the random index (RI), the weights were accepted. The final vulnerability index for the Yucatan case was then defined by Eq. 1:

$$Index = (K \ x \ 0.480) + (VZ \ x \ 0.405) + (S \ x \ 0.115) , \tag{1}$$

where K, VZ and S correspond to the karstification, vadose zone, and soils map layers respectively; numbers inside parentheses are the weights calculated from the AHP.

**4.2 IKAV-A**

For the development of IKAV-A, a previously presented solute transport model was slightly modified. The conceptual model's setup defines the main urban area from each municipality as an infiltration basin for pollution (Martínez-Salvador et al., 2019). Four large well fields (JAPAY I to IV) supplying water for Merida city were defined as stressors of the aquifer. Piezometric data was obtained from six monitoring wells from the metropolitan monitoring network to serve for the calibration process. Precipitation values for monthly and yearly averages were obtained from historic datasets publicly available from the CLICOM-CICESE website (SMN, 2017). As boundary conditions, recharge was estimated from precipitation by the application of the APLIS methodology (Martínez-Salvador, 2018); recharge was sub-divided as: coastal, metropolitan, and

rest of the area. Neumann conditions were settled for eastern and western boundaries and Dirichlet conditions for the southern and northern (sea level) boundaries (Figure 7).

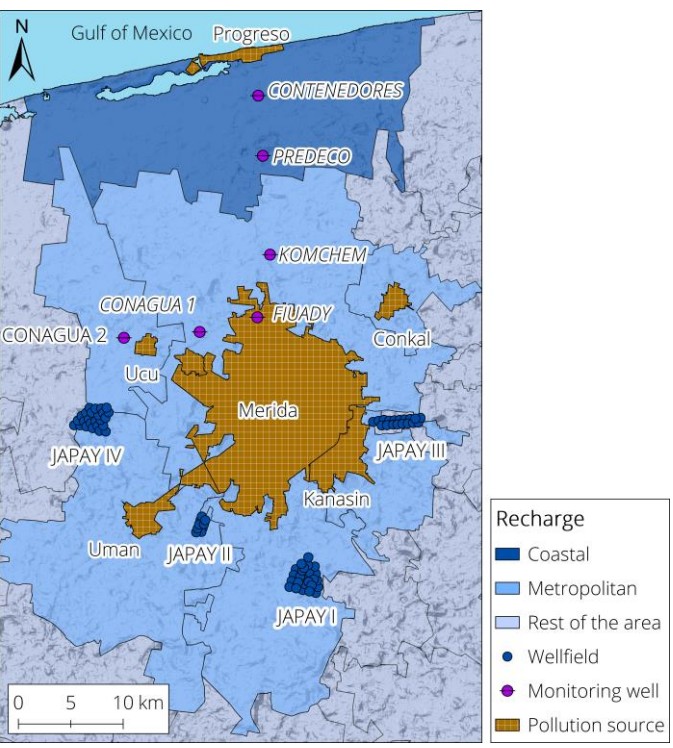

Figure 7: Conceptual model of the MMA.

The sub-surface was defined with four layers, one of them representing a preferential flow layer; this was defined according to drilling works for monitoring wells indicating cavities (Figure 8). The model was run in MODFLOW 2005 (Harbaugh, 2005) utilizing the graphical user interface (GUI) ModelMuse version 3.10.0.0 (Winston, 2009).

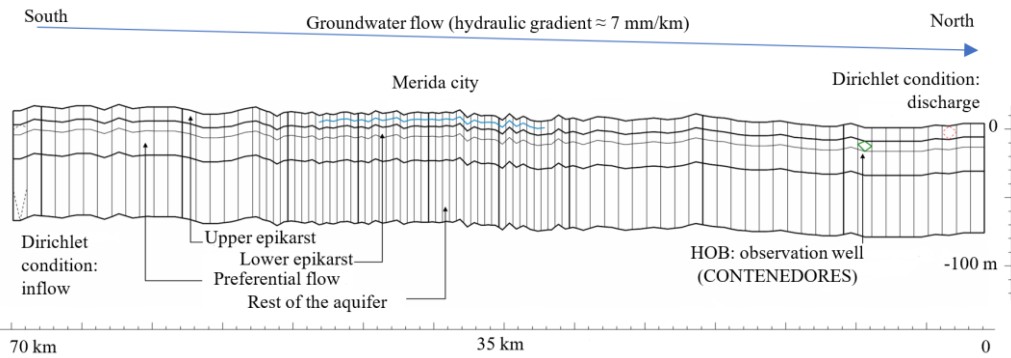

Figure 8: Layer discretization for the MMA subsurface. Modified after Martínez-Salvador et al. (2019).

In order to simulate the turbulent conditions of the preferential layer, the Conduit Flow Process (CFP) package for MODFLOW 2005 was utilized in mode 2 (Shoemaker et al., 2008). After calibration, horizontal hydraulic conductivities (K$x$ and K$y$) were

defined as 1, 0.5, 1.115 and 1 m/s for the up to bottom layers; vertical hydraulic conductivity ($Kz$) was defined homogeneously

as 1.115 m/s. The multi-species solute transport model MT3DS was utilized to simulate advection and dispersion (Zheng and Wang, 1999); given the anthropogenic impact in the MMA, $NO_3^-$ was selected as species. The initial $NO_3^-$ concentration was settled as 80 mg/L in order to approximate previously reported concentrations in the Merida sub-surface (Pacheco-Ávila et al., 2004a; Rojas-Fabro et al., 2015). The contaminant load was settled as continuous for the major urban settlements of the MMA as an attempt to study the effect of the constant leaking of wastewater from septic tanks that actually occurs in the MMA. Additionally, 130 extraction wells were included solely as representative given the lack of data regarding their extraction

volumes and operational times. For more detailed information regarding the model setup, considerations, and calibration the work of Martínez-Salvador et al. (2019) is highly recommended.

In order to define degrees of vulnerability, the permissible maximum of $NO_3^-$ for drinking water was used as a base; according to Mexican standards, this value is settled as 45 mg/L (Diario Oficial de la Federación, 2000). Taking this as a base to determine highly vulnerable water sources, the subsequent vulnerability classes were defined according to previous water quality

monitoring campaigns performed in Yucatan (Pacheco and Cabrera, 1997; Pacheco, 2003; Pacheco-Ávila et al., 2004b; Pérez-Ceballos, 2004; Rojas-Fabro et al., 2015). Given that $NO_3^-$ concentrations can occur naturally, concentrations below 9 mg/L were defined as a low vulnerable condition (Table 3). Although natural occurrence of $NO_3^-$ is dependent on characteristics such as geology and soils, the proposed value for low vulnerability approximates the estimates for non-inhabited areas in Yucatan as presented by Pacheco and Cabrera (1997).

**Table 3: Proposed classification for IKAV-A in the MMA case.**

| Nitrate concentration classification | |
| --- | --- |
| mg/L | Actual vulnerability |
| 0* | Very low |
| 1 to 9 | Low |
| 10 to 22 | Moderate |
| 23 to 45 | High |
| > 45 | Very high |

\* Indicate no influence of the pollution source.

## 5 Results and discussion

### 5.1 IKAV-P

The IKAV-P map indicates that areas of high doline density, suggesting advanced karstification, are the most vulnerable under an incidental pollution scenario in agreement to European methods previously tested in Yucatan (see Moreno-Gómez et al.

(2018), (2019)). However, IKAV-P also classifies the near coastal areas for very high vulnerability (VHV) where very shallow water tables, fissuring, and sands are present. Despite karstification representing the most influential characteristic, given its assigned weight, a distinctive arrangement of some characteristics (coarse soils and a shallow water table) also indicates a

VHV regional condition. A vulnerability reduction pattern, from high vulnerability (HV) to moderate vulnerability (MV), is displayed in relation to an increment of the vadose zone in the Inner Cenote Ring and the Central Plain areas; in the Valleys

and Hills region, the considerable depth of the unsaturated zone promoted a low vulnerability (LV) class. This manifests a consistent interpretation of this coastal aquifer in relation to the contrasting conditions with respect to the flat plain and the southern hill area; both are important regional conditions that were not highlighted by the previously applied intrinsic groundwater vulnerability methods (see Appendix C).

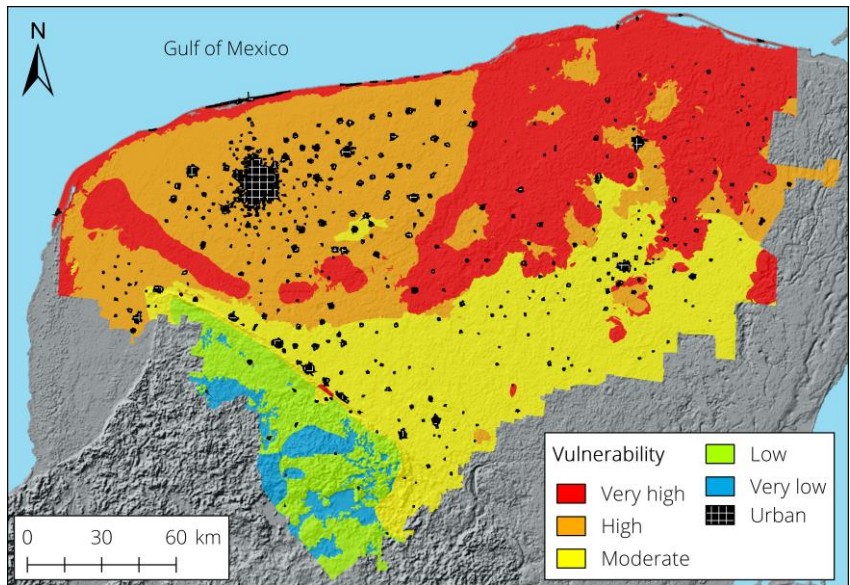

**Figure 9: The IKAV-P map.**

Despite the weight assigned to the soil map being the lowest, it exhibits an important role under different conditions. Soils are depicted as protective when its clay content is above 30%. In the southern area of the Inner Cenote Ring, a clay-rich soil promotes said area for moderate vulnerability, despite the shallow water table and fissuring. Similarly, in the south-eastern Central Plain, clay-rich soils provide some protection in areas were karstification is at its maximum. Soils are the main

promotor for very low vulnerability (VLV) in the Valleys and Hills area where karstification is low and the water table is found at its deepest. In general, results from IKAV-P display percentual vulnerability classes as 30.5, 32.1, 27.2, 7 and 3.3 for VHV, HV, MV, LV, and VLV, respectively.

**4.2. IKAV-A**

Results from the solute transport model indicate that pollution moves northward, following the regional groundwater flow.

Due to the fact that Merida city is the most populated urban area, representing an elevated number of septic tanks leaking wastewater into the aquifer, the effect of the plume in the northern area of the city is more pronounced in comparison with other urban areas. Model layers 1 and 2 (upper and lower epikarst) present similarities on the spatial distribution of $NO_3^-$ concentration, however, model layer 2 is more representative for the IKAV-A source vulnerability given the large number of

extraction wells located at this depth (Figure 10a). The constant leaking of wastewater from septic tanks has already affected
the underlying aquifer in the MMA, however, results indicate that $NO_3^-$ concentrations decrease with depth, not representing
an immediate threat for extraction wells fields located at these depths (Figure 10b and Figure 10c).

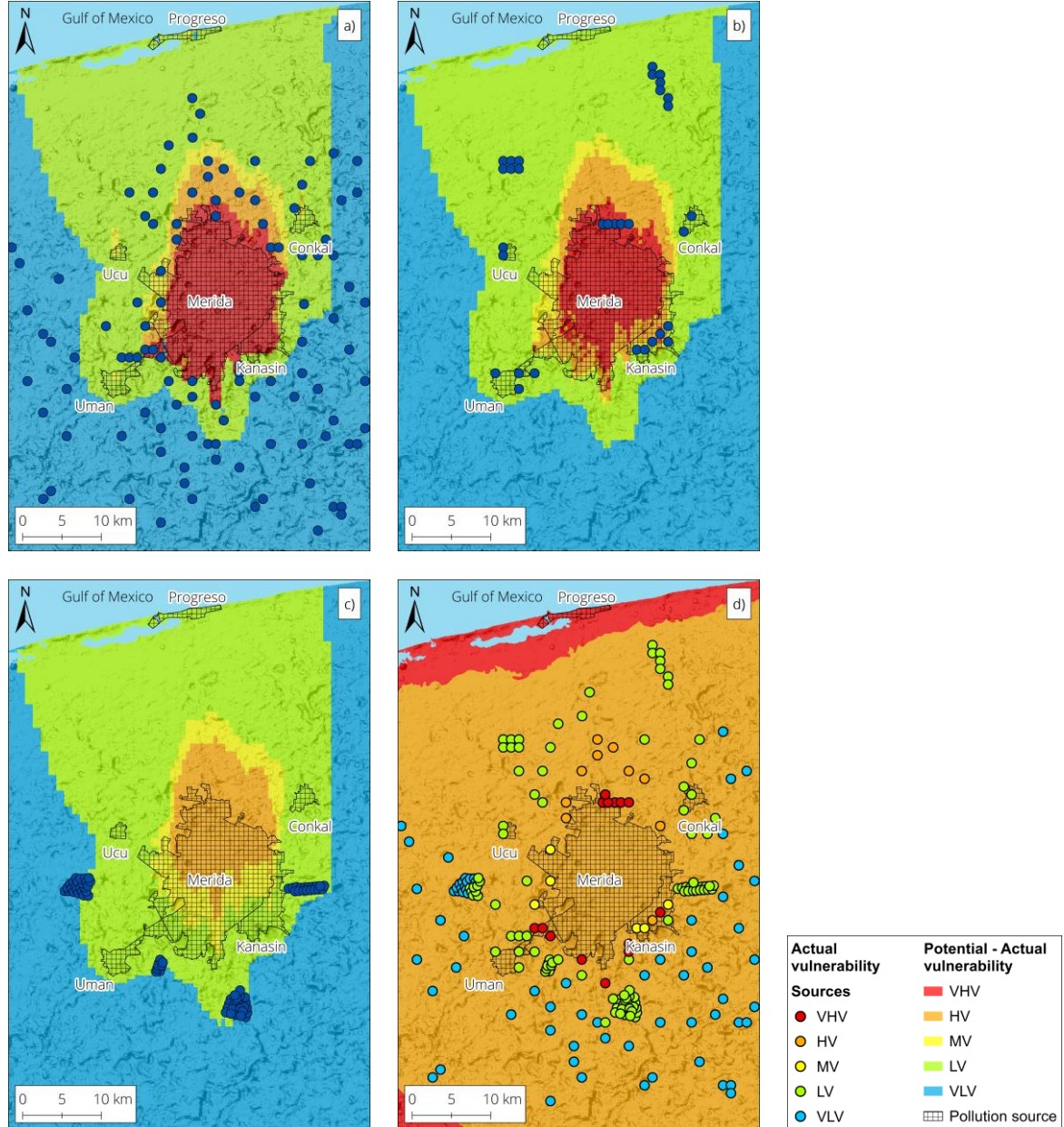

**Figure 10: IKAV-A source vulnerability maps for a 60 years simulation (13th stress period). In a), b) and c), the IKAV-A for the lower epikarst, preferential, and rest of the aquifer layers, respectively; in d), the combined IKAV-P (resource) and IKAV-A (source)**
**for the MMA.**

Comparing results from IKAV-P for the MMA, it is can be clearly seen that no further discretization of vulnerability is possible from the intrinsic characteristics of Yucatan and the data spatial resolution (Figure 10d). Nevertheless, the IKAV-A provides important insights and an additional parameter (pollutant concentration) to expand the analysis in order to improve decisions regarding protection and corrective strategies (Table 4).

**Table 4. Actual vulnerability of extraction wells for water supply.**

| Actual vulnerability (source) | | |
|---|---|---|
| Vulnerability class | Wells | Percentage |
| Very high vulnerability | 14 | 6.8 |
| High vulnerability | 10 | 4.8 |
| Moderate vulnerability | 6 | 2.8 |
| Low vulnerability | 109 | 52.4 |
| Very low vulnerability | 69 | 33.2 |

Although approximately 85% of the extraction wells are classified as low vulnerable, 11% could be severely affected by $NO_3^-$ concentrations higher than the permissible maximum for drinking water. According to the estimated number of inhabitants, who receive water supply from these highly vulnerable wells, approximately 10% of the MMA population ($\approx$ 100,000 inhabitants) can be considered under risk given the simulated $NO_3^-$ concentrations at these extraction points.

Due to the lack of tracer test data, IKAV-P was compared by spatial correlation with the regional intrinsic vulnerability method IVAKY (Aguilar-Duarte et al., 2016). Given that IVAKY categorizes vulnerability into six classes (the additional class is "Extreme vulnerability"), this class was merged with the VHV category for comparative purposes (Figure 11a). Despite the considerable differences in the number of used parameters, attributes, rating schemes, and assigned weights, IKAV-P and IVAKY show a very similar percentual tendency to classify vulnerability in the Yucatan karst (Figure 11b). The percentual similarity is not a definitive indication of spatial relationship, therefore, to investigate the spatial correlation in vulnerability classes between IVAKY and IKAV-P an overlapping process was carried out utilizing ArcGis tools. A remarkable total correlation, above 50%, is displayed by IKAV-P and IVAKY; this correspondence in the assignation of vulnerabilities is outstanding given the fact that the best correlated European methods displayed less than 30% of agreement with the IVAKY method (see maps in appendix C). With these results, the plausibility of the IKAV-P method, to display potential vulnerability in accordance with the principles of regionalization and infiltration scenario, was demonstrated.

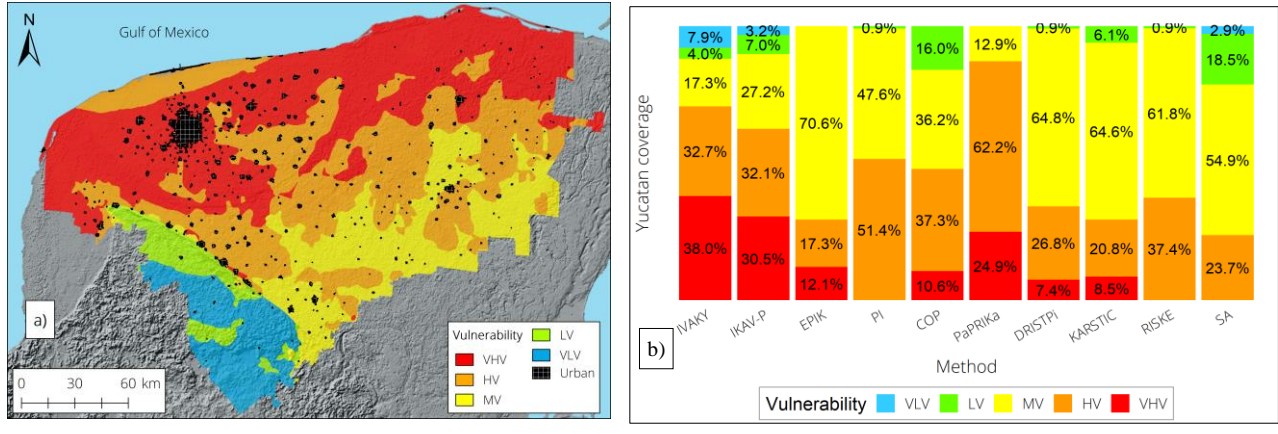

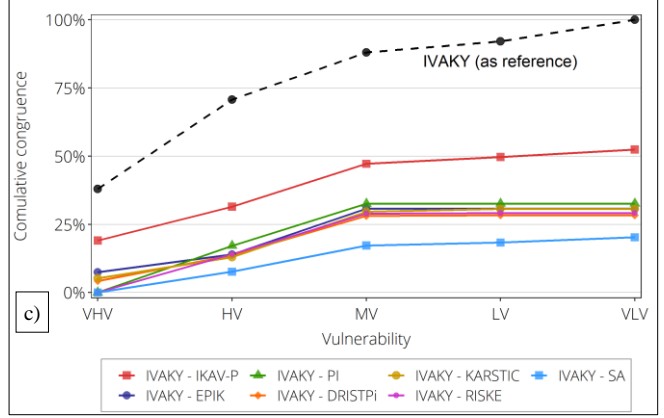

**Figure 11: Validation of IKAP-P. In a), the intrinsic groundwater vulnerability map from the method IVAKY, digitalized from Aguilar-Duarte et al. (2016). In b), comparative of stacked vulnerability percentages with IKAV-P and European methods previously applied in Yucatan; in c), the cumulative spatial correlation of vulnerability from IKAV-P and other methods previously applied in Yucatan.**

Unfortunately, no water quality data was available to validate outcomes from IKAV-A, however, results are consistent with previous studies and water sampling campaigns that took place in Merida and peripheral shallow wells; outcomes from the IKAV-A are, to some degree, consistent with the previous studies (see Pacheco et al. (2001), Pacheco-Ávila et al. (2004a), Pacheco (2004), and Rojas-Fabro et al. (2015), among others). In general, the Merida sub-surface experiences a continuous pollution condition, in which the upper aquifer layers seem to have a permanently high $NO_3^-$ concentration. The pollution plume generated in Merida moves northward according to the natural groundwater flow, temporarily increasing $NO_3^-$ levels in northern areas during the high-rainfall season. In the case of other cities, the pollution seems to be locally generated, with highly transient $NO_3^-$ concentration levels; the temporal $NO_3^-$ variability, reported by preceding water sampling campaigns, is also demonstrated by the model. The increment of $NO_3^-$ levels can be associated with a flush effect, incrementing the pollution in the north of Merida, therefore, the seasonal variations of the representative pollutant should also be considered in further protection strategies.

The empirical interpretation of IKAV-P showed a better approximation of the natural conditions in the Yucatan karst when compared with results from well-established European methodologies previously applied in the same area. By following the steps proposed by the IKAV method, it was possible to obtain a representative potential vulnerability map, highlighting the regional characteristics and their influence on the migration of accidental pollution from the surface to the water table. Regionally, a combination of sands, shallow water table and fracturing also represent a VHV condition, despite the infiltration

scenario in Yucatan settled as diffuse. On the other hand, the most protective natural characteristics in the region are those found in the Valleys and Hills area where high clay content soils, the deepest groundwater table, and a low fracturing are the dominant natural characteristics. It is important to note, that these contrasting conditions were not highlighted by any of the previous European methods applied in Yucatan.

IKAV provides a general guideline to estimate groundwater vulnerability from a regional point of view, but allows for a

necessary flexibility to fulfil the principles and the criteria presented previously. Flexibility is always necessary since interpretations can vary according to data, infiltration scenario, pollutant type, and objective. The combined evaluation of potential and actual vulnerability is demonstrated to be necessary for vulnerability as a decision support tool for karst areas already affected by anthropogenic practices. Therefore, the IKAV can provide an enhanced analysis, beneficial for decision support and the development of strategies for protective, preventative, and corrective measures.

**6 Conclusion**

Given the necessity to minimize the subjectivity of current intrinsic karst groundwater vulnerability methods and to expand the analysis of vulnerability including the anthropogenic influence, the IKAV method is proposed. Following a defined criterion for map layer selection and filtering, a correlative rating system, and a set of principles to be fulfilled, a potential (intrinsic) vulnerability map can be obtained; this procedure aims to minimize the subjectivity and to present a vulnerability

map in accordance with the characteristics of the region being evaluated. By quantifying solute transport emerging from anthropogenic activities, the actual vulnerability of the aquifer (or water sources) can be estimated, expanding the decision scope of vulnerability maps as a decision support tool. Some important contributions of the IKAV for vulnerability studies are understood to be as follows:

• The over-/under-estimation of the vulnerability outcomes is reduced by performing individual analyses for point and diffuse infiltration conditions (infiltration distinctive principle).

• Independent of the number of parameters or attributes, a correlative rating system is favourable for further vulnerability classification.


• The vulnerability index partition is not discretional, but dependent on statistical distribution, allowing a better representation of the study area's characteristics (regionalization principle).

• Vulnerability results are more consistent with regional characteristics when the attributes display a pronounced variability (regionalization principle).

• The evaluation of the anthropogenic influence via solute transport enhances the vulnerability analysis, depicting conditions not displayed by intrinsic methods (representative pollutant principle).

• Actual vulnerability maps can represent individual aquifer layers, providing additional criteria for cost/benefit judgements.

• The IKAV method is not only a vulnerability indicator, it is capable of revealing possible solutions for endangered water sources.

In conclusion, the IKAV method is an improved scheme to estimate potential groundwater vulnerability, integrating solute transport to evaluate an aquifer's current state of vulnerability. The IKAV method reduces the inevitable subjectivity of other vulnerability methods by proposing a workflow with well-defined principles, rules, and systemic attributes evaluation criteria. The IKAV-P map provides decisive insights for protective-preventive procedures and the IKAV-A map focuses on presently vulnerable sections of the aquifer in order to implement corrective actions and maintain optimal groundwater quality. The IKAV was developed considering how parameters, attributes, and values can be assigned to highlight regional intrinsic differences according to infiltration scenarios (regionalization and infiltration distinctive rules). IKAV could be applied for different karst areas inasmuch as the proposed steps and considerations to develop vulnerability maps are followed. This integrated methodology can be taken as the groundwork to expand further vulnerability studies and their role as a decision support tools.

Author Contributions: Conceptualization, M.M-G.; methodology, M.M-G.; validation, M.M-G.; formal analysis, M.M-G. and C.M-S.; investigation, M.M-G.; data curation, M.M-G.; writing—original draft preparation, M.M-G; writing—review and editing, M.M-G., C.M-S., R.L., C.S. and J.P.; visualization, M.M-G. and C.M-S.; supervision, R.L., C.S. and J.P.; funding acquisition, M.M-G.

Funding: This research was funded by CONACYT (CVU: 466945) and the Graduate Academy of the TU Dresden.

Acknowledgments: We acknowledge support by the Open Access Publishing Funds of the SLUB/TU Dresden.

Conflicts of Interest: The authors declare no conflict of interest. The funders had no role in the design of the study; in the collection, analysis, or interpretation of data; in the writing of the manuscript, or in the decision to publish the results.

## Appendix A

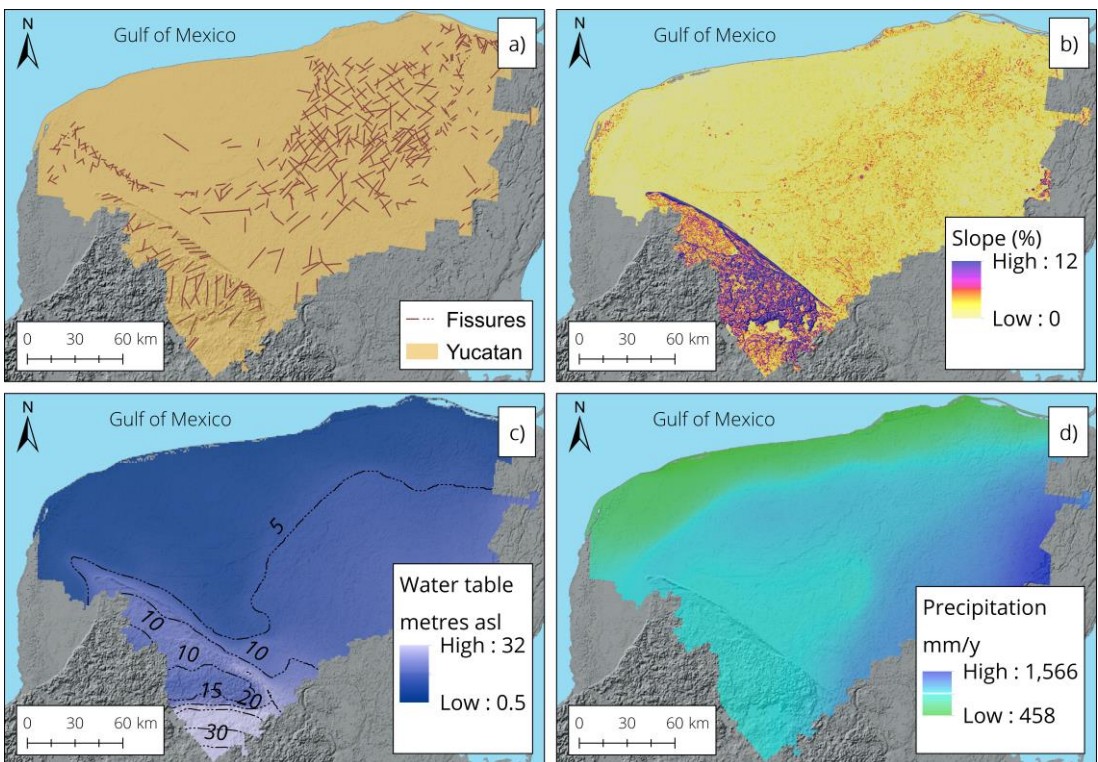

**Figure A1: Regional characteristics of the Yucatan karst. In a), major fissures; in b), the nearly flat conditions of the central plain;**
**in c), the shallow water tables for most of the state and, d), the spatial variability of precipitation. Fissuring map generated from geological datasets from INEGI, (1984); Slope map generated from the digital elevation model ASTER-GDEM version 3, resolution of 30 m from NASA, METI, AIST, Japan Spacesystems and U.S./Japan ASTER Science Team, (2019); Water table elevation map generated from digitalized contour maps from SARH, (1989); precipitation map generated from precipitation data from the CLICOM web platform SMN (2017).**



**Appendix B**

Table B1: Data gathered for the development of previous vulnerability maps in Yucatan. Reproduced from Moreno-Gómez et al. (2019).

| Map layers | DRISTPi | KARSTIC | RISKE | SA | PaPRIKa | COP | PI | EPIK | Base data |
|---|---|---|---|---|---|---|---|---|---|
| Soil thickness | | | X | X | X | X | X | X | Borehole data [a] |
| Soil texture | X | X | X | X | X | X | X | | Borehole data [a]; edaphology maps [a] |
| Lithology | X | X | X | X | X | X | X | | Lithology maps [a] |
| Fracturing | X | X | X | X | X | X | X | | Fissures map [a] |
| Unsaturated zone (depth) | X | X | | X | X | X | X | | DEM [b]; water table [a] |
| Epikarst | | X | | | X | | X | X | Dolines map [a] |
| Confinement | | | | X | | X | | | Literature review |
| Slope | X | X | X | X | X | X | X | X | DEM [b] |
| Vegetation | | | | X | | X | X | X | Land use and vegetation maps [a] |
| Karstification | | X | X | X | X | X | X | X | Dolines/fissure maps [a] |
| Rainfall volume | | | | X | | X | | | 25 years historic data [c] |
| Rainfall intensity | | | | X | | X | | | 25 years historic data [c] |
| Recharge | X | X | | | | | X | | Precipitation [c] |
| Surface features | | | | | | | X | | Dolines and fissures maps [a] |
| Effective field capacity | | | | | | | X | | Settled with minimum values |
| Hydraulic conductivity (soil) | | | | | | | X | | Borehole data [a], Saxton equations |
| Hydraulic Conductivity (aquifer) | | X | | | | | | | No applicable |
| Rock reservoir | | X | X | | X | | | | No applicable |

[a] Data publicly available at http://www.beta.inegi.org.mx/  (Vector maps at 1:50,000 scale).

[b] Digital Elevation Model (ASTER GDEM, 30 metres resolution) from https://www.earthexplorer.usgs.gov/.

[c] Data publicly available at http://clicom-mex.cicese.mx/.


**Appendix C**

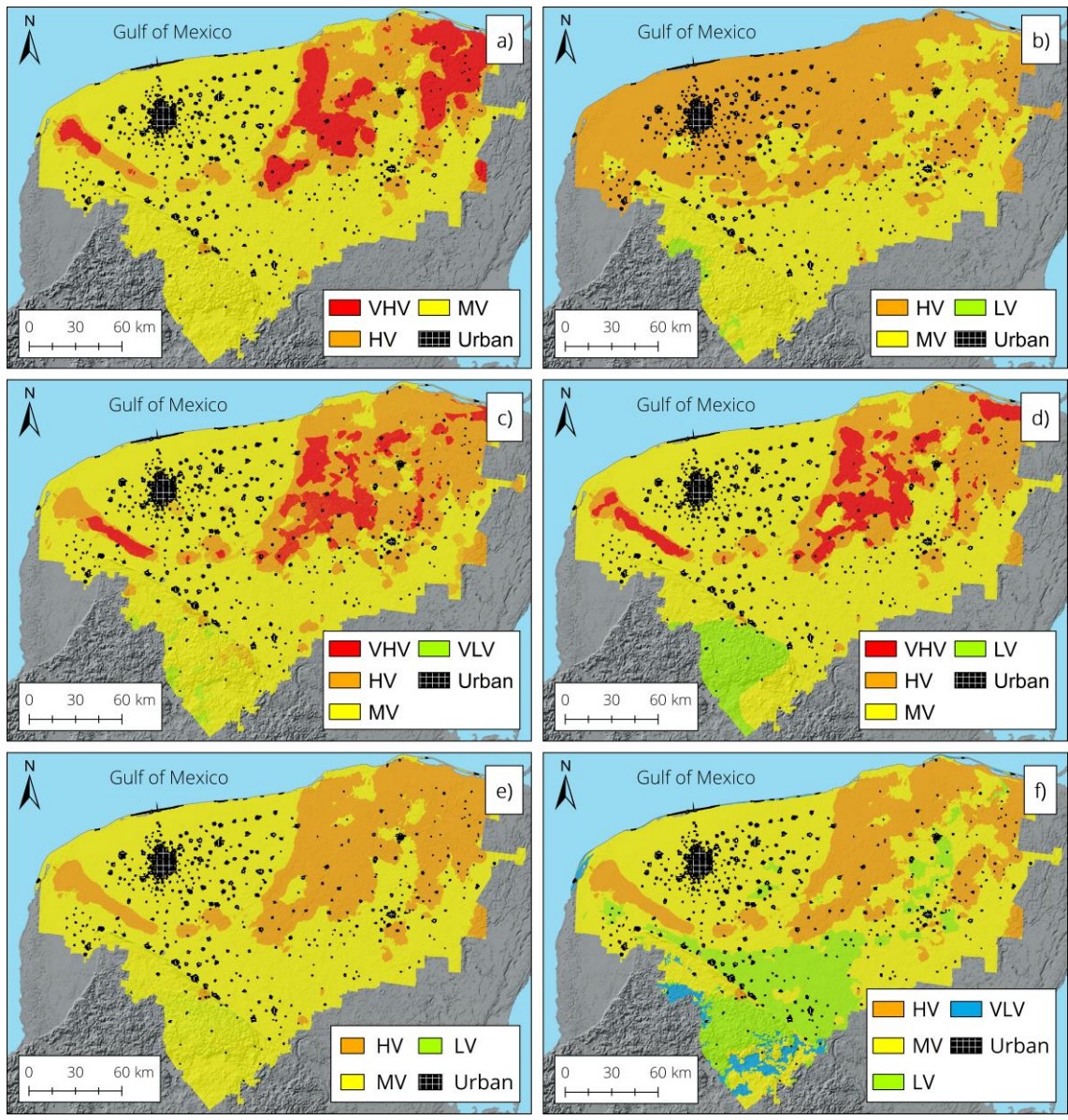


**Figure C1: Vulnerability maps from methods previously applied in Yucatan. From a) to f) EPIK, PI, DRISTPi, KARSTIC, RISKE, and the Slovene Approach respectively. Maps reproduced from Moreno-Gómez (2021).**

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
