# Peer review of "First Application of the Integrated Karst Aquifer Vulnerability (IKAV) method. Potential and Actual vulnerability in Yucatan, Mexico"

_Natural Hazards and Earth System Sciences, 2021_

## Author Response (AR1)

Thank you so much for your comments and suggestions. The improvements you have suggested for the manuscript will be implemented. The following are brief answers to your comments to be follow.

**From Reviewer 1**

1.- Doline and Elevation, as features characteristic maps, were added as Figure 2b and figure 2c.

2.- Similarly, hydrogeological regions were modified to address these regions from maps included in Figure 2b.

3.- Font size in Figure 5 was increased.

4.- Map in Figure 9 was increased as suggested.

5.- A table for results (Table 4) was included as suggested.

**From reviewer 2**

**Broad comments**

**"One of the authors' goals was to reduce the subjectivity of the vulnerability assessment process (abstract, last sentence). However, the process of preparing IKAV-P map does not differ significantly from other PCSM methods with weighting parameters".**

**Answer:** IKAV-P aims to reduce subjectivity from a "regional" point of view. Literature review and results from Yucatan showed that the application of well-established vulnerability methods (e.g., EPIK. COP, PI, RISKE, etc.) does not always highlight important characteristics of studied regions. Despite the process to prepare the IKAV-P map is similar from previous PCSM methods, we propose a series of principles and criteria to be followed in order to obtain vulnerability maps congruent with the regional characteristics where it is applied.

**"Moreover, some parameters are excluded (e.g., Slope) because of low variability, and this approach does not reduce the subjectivity. The final vulnerability map doesn't have to have significant differences particularly if the input parameters do not differ spatially.**

**Answer:** Studies related to multi-criteria decision analysis (e.g., vulnerability, suitability, flood risk, etc.) highlight that as the number of evaluated parameters increase, so it does subjectivity when evaluation is qualitative-based. With this in mind and considering the basic purpose of a vulnerability map (differentiate areas in relation to a predefined process) we suggest the exclusion of homogeneous or low-variable parameters given that such parameter will not contribute to such purpose, minimizing, to some degree, subjectivity.

**"Therefore, specific parameters should be included in the vulnerability assessment process, particularly if two areas are compared (for example, with different terrain morphology)".**

**Answer:** That is correct. To compare vulnerability from one area to another some parameters could be significant for that specific purpose. However, to regionally define vulnerability, some parameters presenting the aforementioned characteristics can be excluded.

**"It is interesting that the precipitation (or infiltration) parameter is not included in the assessment process, although the A1 (Appendix) shows a significant range (458-1566 mm) which could be of significant impact (as previously shown by PI, COP and other European approach methods)".**

**Answer:** Indeed. Methods like COP and the Slovene Approach consider precipitation as an external factor influencing vulnerability. Others such as KARSTIC, DRISTPi or PI consider recharge from precipitation. However, the theoretical approach from these methods considers that "the higher the precipitation/recharge, the more vulnerable the area could be". We consider that high precipitation as direct indicator of high vulnerability can lead to erroneous assumptions in some cases. For example: are areas covered with thick fine-textured soils with 600 mm of rain more vulnerable than those covered with tick sands with 400 mm of rain? Also, COP considers high precipitation, after a preestablished value, as less vulnerable for promoting dissolution. However, to take this approach the volume of solutes for a given area should be known. Having in mind the application of a model including recharge from precipitation for the IKAV-A, it was decided not to include this parameter as part of the IKAV-P.

**"In the bottom line, it should be emphasized that the proposed IKAV method is suitable only for Yucatan and other similar karst environments and not for complex karst systems with allogenic recharge".**

**Answer:** The process to obtain a vulnerability map from the IKAV contemplates a series of steps, filters, and rules to achieve a map in congruence with the characteristics of a given study area. One of these rules is the "infiltration dependent", which contemplates the evaluation of point and diffuse infiltration conditions separately. IKAV can be applied for complex karst systems, including areas promoting allogenic recharge, inasmuch as the proposed rules and steps for parameters selection, attributes definition and values assignation are considered.

**Specific comments**

**"Line 14. Theoretical models that consider contaminant characteristics exist (specific vulnerability, COST 620 report) but are rarely used".**

**Answer:** Yes, specific vulnerability exists but the extensive data and costs for fieldwork is a drawback for its evaluation as mentioned in the COST 620 report. We then proposed modelling to simulate an approximate vulnerable condition.

**"Line 18. Hazard and risk mapping should be mentioned if the anthropogenic impact is introduced into vulnerability assessment".**

**Answer:** You are right. There could be a confusion with actual vulnerability. We have pointed out this.

**"Figure 1. The title should mention that this figure corresponds to PCSM vulnerability assessment methods".**

**"Line 63. It is better to put "in some regions" because the reference Parise et al. 2004 refer only to karst in Albania".**

**"Line 71. Add karst areas where the project "Development of an integrated methodology to estimate groundwater vulnerability to pollution in karst areas" was carried out".**

**"The Study area lacks in description of Quaternary, Neogene and Paleogene units".**

**"Figure 2. Legend does not contain the dark grey geology unit in Tabasco state. It is better to put a blue colour for seas around Mexico on the small map".**

**"Line 93. The four hydrogeological regions are not presented in figure 2 (although presented on other maps)".**

**"Line 97. Reference [29] is unclear".**

**Answer:** We have modified figures, sentences and verified references according the previous seven comments.

**"Line 151. The use of an extensive number of parameters can complicate the process but could be necessary to characterize and evaluate specific karst features".**

**Answer:** We completely agree. What we are proposing is a filtering process to exclude parameters that do not significantly contribute for area discretization in terms of vulnerability. We do not propose neither a maximum nor a minimum of parameters.

**"Line 165. A homogenous layer could be important for groundwater vulnerability, particularly when vulnerability maps of two areas are compared".**

**Answer:** Yes, in terms of comparison with other areas a homogeneous layer can be significant. However, to fulfil the "regionalization" rule homogeneous layers do not contribute to mapping.

**"Line 180. If several pollutants are representative for one area, does it mean that the several IKAV-A maps are to be produced?"**

**Answer:** Unfortunately, yes. Similar to specific vulnerability, IKAV-A can be applied for several pollutants or groups of pollutants with similar characteristics.

**"Chapter 4.1. (IKAV-P). Describe why vegetation and precipitation factors are not used in the IKAV-P assessment (although the precipitation is used in IKAV-A method through recharge)?"**

**Answer:** Methods contemplating vegetation evaluate its role on runoff generation (see EPIK, COP, PI and the Slovene Approach). Given that the study area is nearly flat it was decided not to include this parameter. Although vegetation could play a significant role for recharge, no data regarding spatial distribution of recharge from precipitation is available in the area. Since recharge has been regionally estimated as 20% of precipitation in a regional basis, it was decided to include precipitation as external stressor directly into the model.

**"Line 229. Table 1. If ranges are based on parameters values from the field analysed with the natural breaks' method, then Table 1 presents ranges and ratings only for the Yucatan region. This should be clearly stated in the Table title to avoid using these particular ranges in other study areas".**

**Answer:** You are right, this can create confusion. We will clarify this.

**"Line 264. Are initial $NO_3$ concentrations of 80 mg/l introduced for all pollution source areas presented in figure 7, including smaller towns? Are contaminant loads continuous?"**

**Answer:** Unfortunately, yes. One of the major problems in Yucatan is the lack of sewer systems. The traditional practice is the use of artisanal septic tanks, which are permeable, continuously leaking into the aquifer. The initial nitrate concentration for the model approximates values previously reported by other studies.

**Figure 10 (a, b, c). Which time step represents the modelling plume output?**

**Answer:** Figures correspond to the final time step (13th). The simulation was run for 60 years. This was defined with the aim of approximate current pollution conditions.

**Figure 10d. Wells classified according to IKAV-A vulnerability represent vulnerability which depends on the artificial impact, and therefore this classification represents a risk to contamination. Instead of using source vulnerability, it is better to use actual vulnerability. On the other hand, source vulnerability includes vulnerability assessment through the unsaturated and saturated zones in the groundwater source catchment and could be high without anthropogenic impact. Therefore, it should not be misunderstood with actual vulnerability.**

**Answer:** Exactly, we will try to clarify this statement.

**"Line 323. The presented comparison of IKAV-P with IVAKY and other European methods shows a high spatial correlation with the IVAKY vulnerability map, but it does not mean that IVAKY or IKAV-P maps provide better results than other methods".**

**Answer:** Actually, it does if we theoretically consider, based solely on a regional basis, the most and the less vulnerable conditions in the study area. For example, in Yucatan the most vulnerable characteristics are highly karstified areas (e.g., the doline field), shallow water tables, and coarse-textured soils. On the other hand, the less vulnerable conditions are low karstification, deep water tables (> 100 m) located under the southern hill area, and fine-textured soils. None of the eight applied methods displayed these contrasting characteristics. We will add vulnerability maps from such methods in the Appendix to visually highlight these differences.

**"Conclusion. Considering the previous comments, it should be emphasized that the prosed IKAV method is designed and suitable for Yucatan and other similar karst environments".**

**Answer:** The IKAV was developed considering how parameters, attributes, and values can be assigned to highlight regional intrinsic differences according to infiltration scenarios (regionalization and infiltration distinctive rules). We consider that the IKAV could be applied for different areas inasmuch the steps and considerations to develop a vulnerability map are followed. Currently, IKAV is being evaluated in other areas with contrasting conditions with those from Yucatan. We hope to publish our results soon.

**From Editor**

All comments and suggested modifications for references and text have been modified as requested.